# Characterisation of *Streptococcus suis* Isolates in the Czech Republic Collected from Diseased Pigs in the Years 2018–2022

**DOI:** 10.3390/pathogens12010005

**Published:** 2022-12-20

**Authors:** Monika Zouharová, Bronislav Šimek, Jan Gebauer, Natálie Králová, Ivana Kucharovičová, Hana Plodková, Tomáš Pecka, Marek Brychta, Marie Švejdová, Kateřina Nedbalcová, Katarína Matiašková, Ján Matiašovic

**Affiliations:** 1Veterinary Research Institute, 621 00 Brno, Czech Republic; 2State Veterinary Institute Jihlava, 586 01 Jihlava, Czech Republic; 3Department of Experimental Biology, Faculty of Science, Masaryk University, 602 00 Brno, Czech Republic

**Keywords:** *Streptococcus suis*, serotype, sequence type, pathotype

## Abstract

As in other countries, in the Czech Republic, *Streptococcus suis* infection in pigs is considered an economically significant disease for the pig industry, though little is known about its population structure. We collected *S. suis* isolates from 144 farms in the years 2018–2022. All samples were taken from animals suffering from symptoms indicating possible *S. suis* infection. Serotyping revealed the presence of 23 different serotypes, and 18.94% were non-typable strains. The most common was S7 (14.96%), while other serotypes had frequencies of less than 10%. Sequence typing identified 56 different sequence types, including 31 newly assigned sequence types together with 41 new alleles in genes in the MLST schema. A large portion of isolates (25.70%) were of unknown sequence type. The most common sequence types were ST29 (14.77%) and ST28 (10.04%); the other sequence types had frequencies of less than 10%. In total, 100 different combinations of serotypes and sequence types were identified. Among them, S7ST29 was found in 72 isolates, representing 13.63% of all isolates, and was significantly associated with the central nervous system. Many other isolates of particular serotype and sequence type combinations were found in a few cases, and a number of isolates were non-typable.

## 1. Introduction

*Streptococcus suis* is known as a major swine pathogen that is responsible for significant economic losses in the swine industry. Pathogenic strains are associated with meningitis, arthritis, endocarditis, polyserositis, and septicaemia in piglets and growing pigs [1,2]. Some pathogenic *S. suis* strains have zoonotic potential, causing meningitis in humans [3]. Pigs can be carriers of commensal strains that colonise the upper respiratory tract. Some strains can appear as opportunistic pathogens associated with coinfection with other bacterial or viral pathogens, but if *S. suis* is isolated from the tissue, indicating systemic infection (brain, meninges, heart, and joints), it is considered the primary pathogen [1,4,5].

The *S. suis* population is highly heterogeneous, as different serotypes, phenotypes, and genotypes have been found [6]. Serotyping is the most basic method used to distinguish strains. Currently, there are 29 *S. suis* serotypes [7], which have been described based on serological reactions against capsular polysaccharide (CPS). Methods traditionally used for serotyping are the agglutination and coagglutination tests using serotype-specific antisera [8,9]. However, these serological methods are expensive, time-consuming, and not always clearly readable [10]. Recently, PCR methods targeting the *cps* locus, enabling the rapid serotyping of *S. suis* strains, have been developed [11,12]. However, with standard serological testing or with later-developed specific PCRs, serotype 2 cannot be clearly distinguished from serotype 1/2 and serotype 1 cannot be distinguished from serotype 14 due to the similar genetic contents of their *cps* loci and their serological cross-reactivity [10,13,14]. Differentiation of these serotypes is very important, because serotypes 2 and 14 are associated with human diseases [3]. Additional methods, such as high-resolution melting analysis [15] or a simple and low-cost method for detecting *cpsK* gene polymorphisms based on PCR restriction fragment length polymorphisms (PCR-RFLPs) [16], must be applied to distinguish the mentioned serotypes. Moreover, a whole-genome sequencing (WGS) technique can be used to accurately identify strains [17]. The predominant *S.suis* serotypes isolated from clinical cases in pigs are serotypes 1/2, 1, 2, 6, 7, 9, and 14 [15,18,19], notably in Europe. Besides these serotypes, serotypes 3 and 8 are frequently isolated in North America [20,21]. However, a relatively high percentage of *S. suis* isolates remain non-typable [15,19]. It was shown that more serotypes could be isolated simultaneously in one animal and that different genotypes of the same serotype could be isolated at one timepoint from the same animal [22].

Multilocus sequence typing (MLST) is a method commonly used for the genotyping of pathogens, and numerous sequence types (STs) exist within the *S. suis* species [6]. The MLST method is widely implemented in research to determine the population diversity and global distribution and to understand the epidemiology of *S. suis* [23,24]. The MLST schema, which is widely used today, was originally developed by King et al. in 2002 [23]. The advantage of this method is the database available on the internet (https://pubmlst.org/organisms/streptococcus-suis, accessed on 2 November 2022) which enables the comparison of isolates worldwide, and newly discovered STs from different areas are constantly being added to the database. Global MLST studies of *S. suis* identified ST1, ST25, and ST28 as the most prevalent STs in swine [6,25].

Little is known about the *S. suis* serotypes and sequence types present in the Czech Republic. Previously, information on isolates causing human disorders has been made available [26].

The aim of this study was thus to assess the genetic diversity of *S. suis* strains isolated from diseased pigs in the Czech Republic. The aim was also to determine the associations of serotypes and sequence types with the virulence potentials of the strains specified by the sites of isolation. Due to the possible occurrence of different genotypes of *S. suis* on the same farm or even in the same animal, it is necessary to distinguish virulent strains from less virulent or commensal strains. Identification of the major disease-causing strains could enable the development of specific treatments and effective control plans, including the selection of a vaccine strain.

## 2. Materials and Methods

### 2.1. Isolates

Pigs with suspected presence of *S. suis* were sampled, and *S. suis* isolates were retrieved from organs and body cavity swabs of dead pigs or from nasal swabs of diseased animals. Symptoms indicating possible *S. suis* infection were evaluated by field veterinarians on the farms and were as follows: sudden death, fever, depression, loss of balance, lameness, paralysis, paddling, shaking, and convulsion. Samples from healthy animals were not collected in this study. Some isolates were taken from different organs of the same animal. In total, 528 isolates (Appendix A) were obtained from 144 farms located in the Czech Republic during the years 2018–2022. The data regarding farm identifications were anonymised. In the year 2018, we obtained 104 isolates; we obtained 96 isolates in 2019, 100 isolates in 2020, 146 isolates in 2021, and 82 isolates in the first half of the year 2022.

Blood Agar Base possessing enhanced nutritional properties suitable for the cultivation of *Streptococci* (Blood Agar Base No. 2 CM0271, OXOID, Basingstoke, UK) with 5% defibrinated ram blood (LabMediaServis s.r.o., Jaroměř, the Czech Republic) was used for the primoculture (22–24–48 h, at 37 °C ± 10 °C) of sectional material or swabs. Suspected *S. suis* colonies were further cultivated in microaerophilic conditions (CampyGen, OXOID, Basingstoke, UK) on blood agar plates for 22–24 hours at 37 °C. The pure bacterial culture was tested with a standard biochemical examination using a commercial STREPTOTEST 24 (ERBA LACHEMA, Brno, the Czech Republic), rapid slide latex agglutination (DiaMondiaL Strep Kit; DiaMondiaL, Vienna, Austria), and, finally, the MALDI–TOF method (Matrix-Assisted Laser Desorption/Ionisation–Time of Flight), using a Bruker Microflex mass spectrometer equipped with Maldi Biotyper Compass 4.1.100.10 software, Database RUO 12.0 (updated version) (Bruker, Billerica, MA, USA). The threshold for species identification with a high degree of certainty was log 2.00–3.00, but log 1.70–1.99 was used for species identification with a low degree of certainty. The achieved scores for our investigated isolates were in the range of 1.70–2.6.

### 2.2. Serotyping

Multiplex PCR in four separate PCR reactions [27], with some modifications, was used for serotype determination. The primers targeted the following genes: (i) glycosyltransferase genes *cps1J*, *cps14J*, *cps1/2J*, *cps2J*, *cps3J*, *cps7H*, *cps9H*, *cps16K*, *cps21N*, *cps23I*, and *cps24L*; (ii) capsular polysaccharide repeat unit transporter genes *cps3K*, *cps4M*, and *cps5N*; (iii) the UDP-glucose dehydrogenase gene *cps4N*; (iv) oligosaccharide repeat unit polymerase genes *cps6I*, *cps10M*, *cps11N*, *cps12J*, *cps13L*, *cps15K*, *cps17O*, *cps18N*, *cps19L*, *cps25M*, *cps27K*, *cps28L*, *cps29L*, *cps30I*, and *cps31L*; (v) the N-acetylmannosaminyltransferase gene *cps8H*; and (vi) the glycerophosphotransferase gene *cps25N*.

The first PCR reaction tested the identities of serotypes 1 + 14, 2 + 1/2, 3, 7, 9, 11, 14, and 16 and the presence of the species-specific gene *recN* for *S. suis* verification [28]. *S. suis*-like serotypes (20, 22, 26, 32, 33, and 34) had been recently excluded from the *S. suis* species [28] and could not amplify *recN*. The second reaction tested the identity of serotypes 4, 5, 8, 12, 18, 19, 24, and 25; the third tested serotypes 6, 10, 13, 15, 17, 23, and 31; and the fourth reaction tested serotypes 21, 27, 28, 29, and 30. Polymorphism in the *cpsK* gene was utilised for the further PCR-RFLP resolution of serotype 2 as distinct from 1/2 and 1 as distinct from 14 [16], these serotypes not being distinguishable by multiplex PCR.

Antisera prepared via the immunisation of rabbits with reference strains were used for coagglutination tests [29] for all isolates non-typable by PCR. No positive coagglutination was detected within these isolates.

### 2.3. MLST

Allelic determination of seven housekeeping gene loci (*aroA, cpn, dpr, recA, thrA, gki*, and *mutS*) was performed using multilocus sequence typing (MLST), according to a previously published method [23]. Automatic DNA extraction was performed with the MagNA Pure 96 DNA and Viral NA Small Volume Kit on the MagNA Pure 96 Instrument (Roche Diagnostics International AG, Rotkreuz, Switzerland). PCR products were sequenced by the Sanger sequencing method (Eurofins Genomics, Cologne, Germany) using primers with modifications at the 5′ end to obtain better sequencing results (SEQplus sequencing primers, Generi Biotech, Hradec Králové, Czech Republic).

Identification of alleles and sequence type (ST) assignment were performed using the PubMLST database (https://pubmlst.org/organisms/streptococcus-suis (accessed on 9 November 2022)).

### 2.4. Multiple Correspondence Analysis

Three variables—serotypes, sequence types, and pathotypes (based on the sites of isolation)—were included in multiple correspondence analysis (MCA), processed using R software and R package plugins FactoMineR v.2.4 [30] and factoextra v.1.0.7 [31], both available from the CRAN repository (https://CRAN.R-project.org, accessed on 9 November 2022). Confidence ellipses represent 95% of isolates in each pathotype (the site of isolation).

### 2.5. Association Testing

Associations between pathotypes and serotypes or sequence types, or combinations of serotypes and sequence types, identified among isolates were tested using Pearson’s chi-squared test. Bonferroni corrections for the α level were calculated for each table of combinations (α = 0.05/number of combinations). Due to the very low α calculated, combinations with α < 0.05 were also considered significant.

## 3. Results

### 3.1. Pathotypes

The term pathotype was used for a group of isolates that were isolated from a particular organ or tract. The site of isolation may correspond with pathogenicity. The pathotypes were divided into eight categories:Central nervous system (brain, meninges)—70 isolates (13%);Lower respiratory tract (lungs, broncho-alveolar lavage, swab from chest cavity or pleura)—238 isolates (45%);Upper respiratory tract (tonsils, nasal swab, tracheal swab)—69 isolates (13%);Circulatory and lymphatic systems (pericardium, spleen, lymph nodes)—86 cases (16%);Joints—27 isolates (5%);Gastrointestinal tract (liver, peritoneum, rectal swab, stool)—21 cases (4%);Urogenital system (kidney, bladder, vagina, uterus, amnion, sperm)—13 cases (2,5%);Skin—4 isolates (0,75%);

Isolates from the lower respiratory tract were far more prevalent, representing nearly half of all isolates, followed by isolates from the circulatory and lymphatic systems (16%) and the central nervous system and upper respiratory tract (both 13%) (Appendix A).

### 3.2. Serotypes

Among 528 isolates, 23 different serotypes were identified (Figure 1, Appendix A). Additionally, 100 isolates (18.94%) were not assigned to a known serotype according to the PCR and coagglutination methods. The most frequent serotype was S7, identified in 79 cases (14.96%), followed by S2 (49 isolates, 9.28%), S1/2 (44 isolates, 8.33%), S9 (34 isolates, 6.44%), S8 (29 isolates, 5.49%), S3 (26 isolates, 4.92%), S4 (24 isolates, 4.55%), S1 (22 isolates, 4.17%), S29 (19 isolates, 3.60%), S16 (17 isolates, 3.22%), S31 (17 isolates, 3.22%), S12 (12 isolates, 2.27%), S15 (10 isolates, 1.89%), S5 (9 isolates, 1.70%), S21 (9 isolates, 1.70%), S11 (7 isolates, 1.33%), S19 (6 isolates, 1.14%), S10 (4 isolates, 0.76%), S23 (3 isolates, 0.57%), S13 (2 isolates, 0.38%), S14 (2 isolates, 3.38%), S24 (2 isolates, 0.38%), and S28 (2 isolates, 0.38%). 

Out of 528 isolates, 27 tested RecN-negative (Appendix A). Since all of these isolates were isolated from clinically ill pigs, we decided to include them in the study. Among the 27 RecN-negative isolates, 13 were identified as known serotypes, according to Kerdsin et al. 2014. The identified serotypes were 5, 7, 8, 11, and 12. The remaining 14 isolates were of unknown serotypes. 

### 3.3. Sequence Types

In total, 56 different sequence types were identified, including 31 new STs (Figure 1) submitted to PubMLST (Appendix A). A new ST was assigned when at least two isolates had the same allelic combination. However, 136 isolates (25.70%) were of unknown ST. The most frequent ST was ST29 (78 isolates, 14.77%), followed by ST28 (53 isolates, 10.04%), ST1 (43 isolates, 8.14%), ST54 (28 isolates, 5.30%), ST87 (16 isolates, 3.03%), the newly defined ST2074 (15 isolates, 2.8%), and ST17 (11 isolates, 2.08%). All other STs were identified in fewer than 10 isolates and thus represented less than 2%. 

### 3.4. Relationships between Serotypes and Sequence Types

Within 528 isolates, 100 different combinations of serotypes and sequence types were found (Appendix A). The most prevalent combination of serotype and sequence type was S7 and ST29: out of 79 S7 isolates, 72 were ST29. Similarly, among 44 S1/2 isolates, 30 of them were ST28. Nearly all S1 isolates were ST1 (19 out of 22). S2 isolates were ST28 (22 isolates) and ST1 (17 isolates), with six newly identified as ST2074 and three as ST1805. Among the 29 isolates of S8, 16 were ST87; the other S8 isolates were of different STs. Not surprisingly, the least well-known STs were noted among isolates without serotype determinations.

### 3.5. New Alleles within MLST Genes

Among previously unidentified alleles in MLST genes, 41 were identified in at least two different isolates. These alleles were submitted to the *S. suis* MLST database (https://pubmlst.org/ssuis, accessed on 11 November 2022) and received assignments, as shown in Appendix A.

### 3.6. New Sequence Types

New sequence types were identified by new combinations of previously known MLST gene alleles or they included newly identified alleles. A new ST was assigned when it was found in at least two different isolates. In total, 31 new STs were identified and deposited in the PubMLST database (Appendix A).

### 3.7. Associations between Pathotypes, Serotypes, and Sequence Types

The possible relationships between serotypes, sequence types, and pathotypes (represented by the site of isolation) were analysed using multiple correspondence analysis (MCA) (Figure 2), which showed the first and second dimensions that represented less than 5% of the data. Pathotypes that could be considered less pathogenic (upper respiratory tract, urogenital system, gastrointestinal tract, and skin) were slightly shifted from other, more pathogenic pathotypes, whose confidence ellipses overlapped and formed a cluster of very similar features (central nervous system, circulatory and lymphatic systems, lower respiratory tract, and joints). 

### 3.8. Associations between Serotypes and Pathotypes

Associations between serotypes and pathotypes are depicted in Figure 3. For 24 different serotypes and 8 different categories of pathotypes, the α level calculated was α = 0.05/192 = 0.00026. Significant associations with α < 0.00026 were found for S7 with the central nervous system and for S21 and S29 with the upper respiratory tract. Associations at α < 0.00026 were also found among the lower respiratory tract and urogenital system with isolates of non-detectable sequence types. Associations at α < 0.05 were found for S8 with the lower respiratory tract, S11 with the upper respiratory tract, S1/2 and S9 with the circulatory and lymphatic systems, and S7 with joints.

### 3.9. Associations between Sequence Types and Pathotypes

As with serotypes and pathotypes, sequence types were also tested for associations with pathotypes (Figure 4). For 57 different STs and 8 different pathotypes identified, the α level calculated was α = 0.05/456 = 0.000109. No associations of STs with pathotypes were significant at this level; however, for the association of ST29 with the central nervous system, the α calculated was 0.00015. Other associations at α < 0.05 were as follows: ST1 was associated with the circulatory and lymphatic systems and with the lower respiratory tract; ST15, ST16, ST28, and ST87 were also associated with the lower respiratory tract; ST29 was associated with joints; and isolates with no detectable ST were associated with the upper respiratory tract.

### 3.10. Associations between Particular Serotype and Sequence Type Combinations

When testing the associations of 100 different combinations of serotypes and sequence types identified in our study with 8 pathotypes, the α level calculated was α = 0.05/800 = 0.0000625. The only association at this α level was found for S7ST29 and the central nervous system. The associations significant at α < 0.05 were for the combinations of S2 and ST28 and S8 and ST87, both with the lower respiratory tract (Appendix A).

## 4. Discussion

The data presented in our study showed high diversity among *S. suis* isolates collected in the Czech Republic from diseased pigs during the years 2018–2022. We identified 23 different serotypes, 56 sequence types, and 100 combinations of serotypes and sequence types. Moreover, a significant proportion of isolates were non-typable with respect to serotypes or sequence types or both.

Isolates from the lower respiratory tract were the most frequent among isolates from organs or systems. These isolates, together with isolates from the upper respiratory tract, gastrointestinal tract, urogenital system, and skin, may not have been the direct cause of death [6]. On the other hand, in multiple correspondence analysis, lower respiratory tract isolates clustered together with isolates from the central nervous system and the circulatory and lymphatic systems, as discussed below. The clinical relevance of this main group of isolates is thus uncertain. Isolates from the central nervous system and the circulatory and lymphatic systems we considered to be highly pathogenic.

Isolates of serotype 7 represented more than 10% of all isolates. Isolates of other serotypes (S1/2, S1, S2, S3, S4, S8, and S9) known to be present in pathogenic strains [6,32] were less frequent but represented at least 3% of all isolates. Ten different serotypes represented up to nine isolates (below 1.8%), but more than 18% of isolates were non-typable for known serotypes using multiplex PCR [27]. This finding demonstrates the high diversity of *S. suis* in the Czech Republic, which is higher than the diversities currently reported in Switzerland [33], North America [6], and China [34], but similar to that reported in Germany [32]. Interestingly, serotype 6, recognised together with serotype 9 as a main isolate serotype (14.8% and 15.9%, respectively) in Switzerland [33], was not identified in any of our isolates.

Similar to previous works [6,33,35], serotype 7 was associated with pathogenicity. In our case, S7 was associated significantly (after Bonferroni correction) with presence in the central nervous system. Serotype 7 thus could be, at least in Europe and North America, considered to be pathogenic, such that attention should be paid to it. Strong associations were found, also, for non-typable isolates with the urogenital system and the lower respiratory tract. This finding is surprising, because the group of non-typable strains should be considered genetically diverse, as 18 different sequence types are documented within this group. Associations significant at α lower than 0.05 but higher than α after Bonferroni correction were also found for isolates of S8 with the lower respiratory tract, S1/2 and S9 with the circulatory and lymphatic systems, and S7 with joints.

Isolates of serotypes 21 and 29 were both associated with the upper respiratory tract and thus should be considered as having low pathogenicity or as being non-pathogenic, despite the fact they were isolated from diseased animals. Serotype 29 was considered a commensal and serotype 21 a commensal or opportunistic pathogen also by Estrada [6]. Moreover, isolates of serotype 11, not detected in Switzerland [33] or North America [6], but detected in China [34] and Germany [32], were associated at α < 0.05 with the upper respiratory system and thus should be considered as having low pathogenicity or as commensal.

In our study, we included 27 isolates that tested RecN-negative and which thus could be considered not *S. suis*, according to typing developed by Ishida [28]. The reason for including them in the study was that they were isolated from clinically ill pigs and could thus be considered pathogenic to pigs. Moreover, 13 RecN-negative isolates were typed for known *S. suis* serotypes. This may open discussion about the specificity of RecN for determining *S. suis* species identity.

Associations of pathotypes with serotypes were more frequent than associations with sequence types or serotype–sequence type combinations. Moreover, the significance of associations of pathotypes with serotypes was stronger, as five different serotype–pathotype associations were lower than the α level after Bonferroni correction. However, this fact was influenced by the very low α level calculated by Bonferroni correction with the increasing number of tests in the case of sequence types and serotype–sequence type combinations. 

Similar to serotypes, only sequence types 29 and 28 were found in more than 10% of isolates. These two sequence types, belonging to the same clonal complex [33], represented nearly one-quarter of all isolates. Moreover, ST29 was significantly associated with the central nervous system, as the only ST within our study, and ST29 isolates currently present in the Czech Republic should thus be considered highly pathogenic, similar to ST29 present in Germany [32]. This finding is in agreement with previous work, where ST29 was associated with pathogenicity also in North America [6]. In contrast to our own and German [32] collections, ST29 was found to be a rare sequence type in North America [6]. 

The ST1 isolates were the only ones associated with the circulatory and lymphatic systems, and ST1 was thus confirmed to possess high pathogenic potential. These findings are in concordance with previous observations [6,33].

Multiple correspondence analysis based on pathotypes, serotypes, and sequence types showed clustering into four main clusters. One cluster was composed of isolates from the upper respiratory tract, the second was composed of isolates from the urogenital system, and the third was composed of isolates from skin and the gastrointestinal system, while the rest of the isolates were from the central nervous system, the circulatory and lymphatic systems, and the lower respiratory tract. Although the authors of a previous work [6] recognised only three categories of pathotypes (commensal, possibly opportunistic, and pathogenic), their clustering was very similar to our results.

Relatively high numbers of serotypes or sequence types were present in only a few isolates. This was particularly apparent for newly described sequence types: we identified 41 new alleles in genes within the MLST schema and 31 new sequence types. A possible explanation of this fact is that a substantial proportion of these isolates were isolated only by chance and were not primary pathogens causing death or major signs of disease. On the other hand, all samples were collected from animals suffering from disease with symptoms common to the *S. suis* course of infection, as reported by field veterinarians. S7ST29 was significantly associated with the central nervous system, similar to S7ST29 isolates in Germany [35]. Due to the similar pathogenicity of our S7ST29 and German S7ST29 isolates, we can speculate that the S7ST29 isolates present in the Czech Republic and Germany may represent the same strain. This possibility is supported, also, by extensive pig transfer between the Czech Republic and Germany [36].

The other sources of diversity are the group of non-determined serotypes and the group of non-determined sequence types. With respect to the non-determined serotype group, it was significantly associated with the lower respiratory tract and also with the urogenital system; we will pay attention to this group in further studies. In addition, while non-determined serotypes have been found by other authors in their works, the size of this group varies among studies. In a recent study in North America from the years 2014–2017, the percentage of isolates of unknown serotypes was 11% [6]; in Switzerland, in the last decade, it was 9% [33]; but in Germany, in the years 2015–2016, it was only 4.2% [32]. On the other hand, in China, in the years 2015–2017, it was 14% [34]. In our work, it was nearly 19%, although 2.6% of non-typable isolates may be other *Streptococcus* species. This high portion of non-typable isolates may be due to the predominance of autochthonous serotypes in Eastern Europe before borders were opened in the late nineties of the last century. There are no studies on serotypes in the Czech Republic in those years, and, to the best of our knowledge, our work is the first ever study on *S. suis* diversity in pigs in the Czech Republic.

Although the information presented in our work is based on data collected in the Czech Republic, we believe that the findings may be of interest to Western European countries due to the high mobility of pigs in the European Union and their being the source of a significant amount of breeding material and also of weaned piglets on Czech farms [36].

## Figures and Tables

**Figure 1 pathogens-12-00005-f001:**
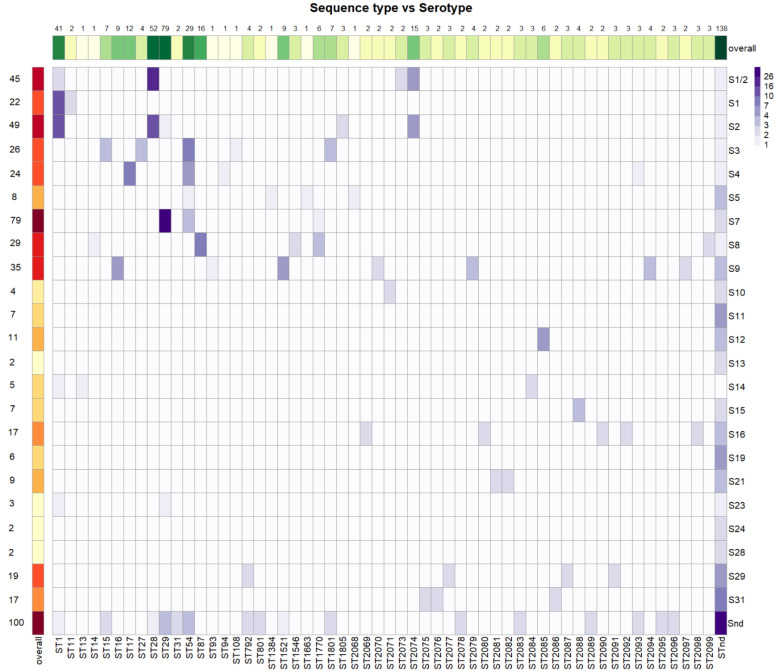
Serotypes and sequence types identified. Snd = serotype not determined, STnd = sequence type not determined.

**Figure 2 pathogens-12-00005-f002:**
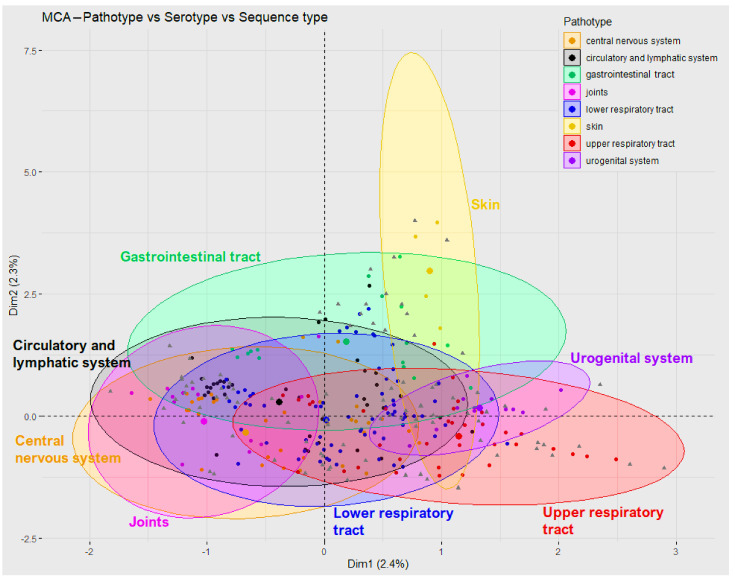
Multiple correspondence analysis. Multiple correspondence analysis of three variables—pathotype (site of isolation), serotype and sequence type. Confidence ellipses represent 95% of isolates in each pathotype. Grey triangles represent the three variables (pathotype, serotype, sequence type); dots represent each isolate and are coloured according to pathotype. The names of isolates and variables are not shown due to a high number of entities.

**Figure 3 pathogens-12-00005-f003:**
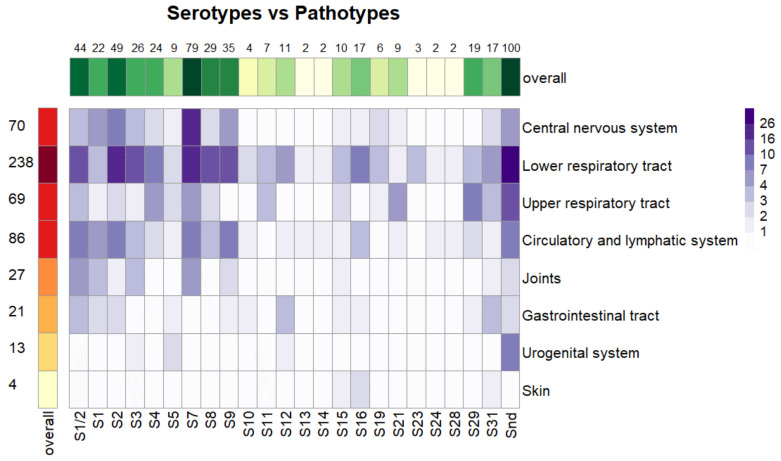
Associations between serotypes and pathotypes. Numbers represent the numbers of isolates within serotypes and pathotypes. Snd = serotype not determined.

**Figure 4 pathogens-12-00005-f004:**
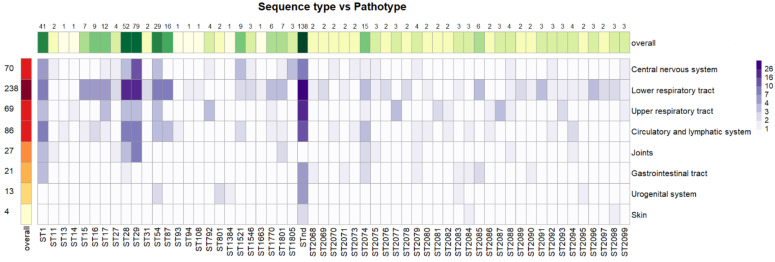
Total counts of different pathotypes within sequence types. Numbers represent the numbers of isolates within sequence types and pathotypes. STnd = sequence type not determined.

## Data Availability

The new allele sequences and sequence types were submitted to the *S. suis* MLST database (https://pubmlst.org/ssuis, accessed on 11 November 2022).

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
