# Peer review of "Characterisation of Streptococcus suis Isolates in the Czech Republic Collected from Diseased Pigs in the Years 2018–2022"

_pathogens, 2022, doi:10.3390/pathogens12010005_

Round 1

Reviewer 1 Report

The main aim of presented manuscript is a characteristic of Streptococcus suis, isolated in Czech Republic. The topic is important and this study should be published, but it needs some upgrades. The work is generally clearly written, but there are some points which need to be corrected. The sentence with the aim of the study (line 79-81) is too long - it needs to be shorter or the aim should be placed in 2 sentences. The Agar Base no. 2 is a trade name, so it should be better to clarify this case - do the Authors mean the CM0271 (by Oxoid) - if yes please give the full information of this product, because it must be clear for the reader and the reader should be able to make a potential replication of this study. Line 159-161 is not clear, please rewrite these sentences.  Line 162-170, please use points or dots. There are also some doubts related to the figures. Figure no. 1 related to pathotypes is too small, there are problems to read the text - it should be corrected. Figure no. 2 is also hard to read and make an interpretation. There is a lot of important informations, so please considerate to place and present these data in another way or in two separate figures. Figure no. 5, 6, 7 - similar comments to mentioned above (an Fig. no. 6 is completely in unacceptable form!).

Generally, besides critical comments, it was a pleasure to read this manuscript, especially because in my opinion the topic is very important and actual as well as there is a strong scientific impact in this paper.

Congratulations to the authors for the well done study 

Author Response

reviewer 1:

The main aim of presented manuscript is a characteristic of Streptococcus suis, isolated in Czech Republic. The topic is important and this study should be published, but it needs some upgrades. The work is generally clearly written, but there are some points which need to be corrected. The sentence with the aim of the study (line 79-81) is too long - it needs to be shorter or the aim should be placed in 2 sentences. The Agar Base no. 2 is a trade name, so it should be better to clarify this case - do the Authors mean the CM0271 (by Oxoid) - if yes please give the full information of this product, because it must be clear for the reader and the reader should be able to make a potential replication of this study. Line 159-161 is not clear, please rewrite these sentences. Line 162-170, please use points or dots. There are also some doubts related to the figures. Figure no. 1 related to pathotypes is too small, there are problems to read the text - it should be corrected. Figure no. 2 is also hard to read and make an interpretation. There is a lot of important informations, so please considerate to place and present these data in another way or in two separate figures. Figure no. 5, 6, 7 - similar comments to mentioned above (an Fig. no. 6 is completely in unacceptable form!).

Generally, besides critical comments, it was a pleasure to read this manuscript, especially because in my opinion the topic is very important and actual as well as there is a strong scientific impact in this paper.

Congratulations to the authors for the well done study 

Response: Thank you very much for valuable comments.

The aim sentence on lines previously numbered 79-81 was divided into two sentences – see lines 81-83.

Thank you for this comment, the agar used was the CM0271 from Oxoid. The details about agar base were amended – see line 104.

We try to clarify the definition of the pathotype term, the sentences on lines originally numbered 159-161 were rewritten (now lines 165-167).

The term pathotype was used for a group of isolates that were isolated from a particular organ or tract. The site of isolation may correspond with pathogenicity. The pathotypes were divided into eight categories:”

Reviewer 2 Report

In this manuscript, authors characterized S. suis strains recovered from diseased pigs from different farms located in Czech Republic between 2018-2022. In total, 528 isolates were characterized based on the isolation site (tissue/organ), serotype and sequence type. In overall, the authors have drawn up a great portrait of the collected strains of S. suis found throughout the targeted farms of the Czech Republic between 2018-2022 and found some interesting associations between serotype, sequence type and isolation site.

Although the article is relatively well written and follows a logical order, I would have some suggestions to improve the quality of the scientific article and for ease of reading/interpretation.

Point to address:

1.     One of my concerns is about the use of the MALDI TOF to confirm S. suis identification. To my knowledge, the ‘gold standard' for S. suis identification is PCR with species-specific gene (recN) and this method was validated. There is little confusion if the authors also used this PCR method to confirm S. suis ID. In M&M (line 114) authors stated that the primers were adopted from a previous study (reference #27). I think it may be the wrong reference (Van Hout J. & al. 2016) because I cannot find any serotyping primers in this paper. If the authors are referring to reference #26 (Kerdsin & al. 2014), there are using the gdh gene to confirm S. suis and cannot distinguish 20, 22, 26, 32, 33, 34 S. suis-like strains. Could authors (1) correct the reference if needed and (2) specify if they used recN to confirm S. suis with the appropriate reference?

2.     As described in this paper, a lot of strains were non-typeable by PCR or/and by the serological reaction. Have these strains been tested and are positive for recN? Please specify. I think it is a critical point to clarify to support the conclusions.

3.     Did the authors use a validated protocol for MALDI TOF (threshold, etc..)

4.     Could the authors introduce some relevant information about S. suis situation in the Czech Republic in the introduction? Prevalent serotype, surveillance program (if any)…

5.     Pathotype mainly refer to the genetic background (virulence factor) and clinical symptom, not the isolation site. The isolation site is not a determinant of the pathogenicity. I would consider changing this term.

6.     Line 162-169; Authors should add this into a table and remove the corresponding figure.

7.     Tables 1 and 2 should be in supplemental materials

8.     I would strongly consider changing the way to present Figs 5 and 7. It is really difficult to interpret, too many colors (for instance, s3 s16 almost identical in fig 6), and there is no quantitative information. Maybe a heatmap?

9.     What is the frequency of NT strains in others studies? Please discuss in the discussion.

10.  Do the authors have a hypothesis of the great diversity of strains in the Czech Republic? Could be added to the discussion.

11.  Lines 402-404; very interesting! How could this concretely affect Western European countries?

Minor points:

-Line 188; and serotyping as well?

-Line 204; a little bit misleading:  56 characterized ST were detected and 31 new or uncharacterized ST were also detected?

Author Response

Reviewer 2:

In this manuscript, authors characterized S. suis strains recovered from diseased pigs from different farms located in Czech Republic between 2018-2022. In total, 528 isolates were characterized based on the isolation site (tissue/organ), serotype and sequence type. In overall, the authors have drawn up a great portrait of the collected strains of S. suis found throughout the targeted farms of the Czech Republic between 2018-2022 and found some interesting associations between serotype, sequence type and isolation site.

Although the article is relatively well written and follows a logical order, I would have some suggestions to improve the quality of the scientific article and for ease of reading/interpretation.

Point to address:

  1. One of my concerns is about the use of the MALDI TOF to confirm S. suis identification. To my knowledge, the ‘gold standard' for S. suis identification is PCR with species-specific gene (recN) and this method was validated. There is little confusion if the authors also used this PCR method to confirm S. suis ID. In M&M (line 114) authors stated that the primers were adopted from a previous study (reference #27). I think it may be the wrong reference (Van Hout J. & al. 2016) because I cannot find any serotyping primers in this paper. If the authors are referring to reference #26 (Kerdsin & al. 2014), there are using the gdh gene to confirm S. suis and cannot distinguish 20, 22, 26, 32, 33, 34 S. suis-like strains. Could authors (1) correct the reference if needed and (2) specify if they used recN to confirm S. suis with the appropriate reference?

Response: Thank you very much for this important comment. We used PCR for RecN detection according to Ishida et al. 2014 in all samples.

We removed the sentence “The primers used for serotyping were adopted from a previous study [27 - Van Hout J. & al. 2016].” on lines previously numbered 114-115, because all the PCR-based serotyping was done according to Kerdsin et al. 2014 (now the same reference number 27).

We apologize for use of incorrect references.

Within 528 isolates, 27 of them were repeatedly tested as RecN negative. Because all these isolates were isolated from clinically ill pigs, we decided to include them in the study. Among 27 RecN negative isolates, 13 of them were identified as known serotypes according to Kerdsin et al. 2014. The identified serotypes were 5, 7, 8, 11 and 12. The remaining 14 isolates were of unknown serotypes.

The RecN negative isolates of unknown serotypes are probably other Streptococci previously marked as suis-like isolates. The identity of RecN negative isolates typed for known S. suis serotype is questionable. We would like to identify them by NGS in the following project.

The Results and Discussion were supplemented with information about RecN negative isolates. Please see lines 197-201 and 347-352 respectively.

2. As described in this paper, a lot of strains were non-typeable by PCR or/and by the serological reaction. Have these strains been tested and are positive for recN? Please specify. I think it is a critical point to clarify to support the conclusions.

Response: Thank you for this comment. We add RecN genotyping results on data Supplement table S1. For details, please see previous response.

3. Did the authors use a validated protocol for MALDI TOF (threshold, etc..)

Response: Thank you for the comment. The thresholds scores ranged from (log) 2.00 to 3.00 for the species identification with a high degree of certainty, score (log) 1.70 - 1.99 for species identification with a low degree of certainty and score log 0.00 - 1.69 for not possible species identification. The achieved score of our investigated isolates was in the range 1.70 - 2.6. We append this information on lines 114-117.

4. Could the authors introduce some relevant information about S. suis situation in the Czech Republic in the introduction? Prevalent serotype, surveillance program (if any)…

Response: Thank you for this comment. To the best of our knowledge, there was no previous surveillance program for S. suis in the Czech Republic. The only available information about S. suis serotypes is for human isolates. In the Introduction, we add this information on lines 79-81.

5. Pathotype mainly refer to the genetic background (virulence factor) and clinical symptom, not the isolation site. The isolation site is not a determinant of the pathogenicity. I would consider changing this term.

Response: Thank for this comment. Although some authors use the term pathotypes for isolates having particular virulence factors (Prufer et al. 2019 https://doi.org/10.1371/journal.pone.0210801), others use the term for some degree of pathogenicity (Estrada et al. 2019 https://doi.org/10.1128/JCM.00377-19.).

In our work we used the term pathotype for isolates from particular organs or systems, similarly to Estrada et al. 2019 but in more detailed resolution. We believe the site of isolation could correspond with pathogenicity. We suppose the site of S. suis isolation from heavily diseased or dead animals should correspond to pathogenicity. For example, only highly pathogenic strains are able to penetrate the brain. This point of view is also consistent with the Wikitionary definition: "Pathotype: Any of a group of organisms (of the same species) that have the same pathogenicity on a specified host." We add the following clarification on lines 165-167: "We use the term pathotype for isolates that have the same pathogenicity. The same pathogenicity is defined by the site of isolation from following organs or group of organs divided to eight categories:"

6. Line 162-169; Authors should add this into a table and remove the corresponding figure.

Response: Thank you for suggestion. The description of pathotypes contains a relatively large amount of information that cannot easily fit into a table. We add bullet points to better separate each pathotype description. The Figure 1 was removed, the same information is available in the next figures, now numbered as the Figure 3 and Figure 4.

7. Tables 1 and 2 should be in supplemental materials

Response: Thank you for this consideration. The tables were removed from manuscript and put in supplemental material as the Supplementary Table S2 and Supplementary Table S3.

8. I would strongly consider changing the way to present Figs 5 and 7. It is really difficult to interpret, too many colors (for instance, s3 s16 almost identical in fig 6), and there is no quantitative information. Maybe a heatmap?

Response: Thank you very much for great idea. We tried to substitute bar graphs with heat maps and found it is much better. Not only does one heat map substitute figures 4 and 5 or 6 and 7, respectively, but also the heat maps are much clearer.

9. What is the frequency of NT strains in others studies? Please discuss in the discussion.

Response: Thank you for this comment. The Discussion for this topic was extended on lines 394-400.

10. Do the authors have a hypothesis of the great diversity of strains in the Czech Republic? Could be added to the discussion.

Response: Thank you for this suggestion. We extend the Discussion for this topic on lines 400-405.

 11. Lines 402-404; very interesting! How could this concretely affect Western European countries?

Response: The breeding material and weaned piglets are transferred from Western Europe into the Czech Republic but rarely in the opposite way. The epidemiological situation in the Czech Republic thus could not affect Western European countries.

Minor points:

-Line 188; and serotyping as well?

Response: Only isolates non typable by PCR method were tested also by co-agglutination method. The information was added to the line 189.

-Line 204; a little bit misleading: 56 characterized ST were detected and 31 new or uncharacterized ST were also detected?

Response: Thank you for this note. In total, 56 STs were found, including 31 new STs submitted to PubMLST database. The sentence on lines 210-211 was corrected to be more clear.